# Towards universal health coverage for people with stroke in South Africa: a scoping review

Sjan-Mari van Niekerk,[1] Sureshkumar Kamalakannan [iD] ,[2,3]
Gakeemah Inglis-Jassiem,[1] Maria Yvonne Charumbira [iD] ,[4] Silke Fernandes,[5]
Jayne Webster,[6] Rene English,[7] Quinette A Louw,[1] Tracey Smythe [iD] [8]

For numbered affiliations see end of article.

**Correspondence to**
Mr Sureshkumar Kamalakannan;
suresh.kumar@lshtm.ac.uk

## ABSTRACT

**Objectives** To explore the opportunities and challenges within the health system to facilitate the achievement of universal health coverage (UHC) for people with stroke (PWS) in South Africa (SA).

**Setting** SA.

**Design** Scoping review.

**Search methods** We conducted a scoping review of opportunities and challenges to achieve UHC for PWS in SA. Global and Africa-specific databases and grey literature were searched in July 2020. We included studies of all designs that described the healthcare system for PWS. Two frameworks, the Health Systems Dynamics Framework and WHO Framework, were used to map data on governance and regulation, resources, service delivery, context, reorientation of care and community engagement. A narrative approach was used to synthesise results.

**Results** Fifty-nine articles were included in the review. Over half (n=31, 52.5%) were conducted in Western Cape province and most (n=41, 69.4%) were conducted in urban areas. Studies evaluated a diverse range of health system categories and various outcomes. The most common reported component was service delivery (n=46, 77.9%), and only four studies (6.7%) evaluated governance and regulation. Service delivery factors for stroke care were frequently reported as poor and compounded by context-related limiting factors. Governance and regulations for stroke care in terms of government support, investment in policy, treatment guidelines, resource distribution and commitment to evidence-based solutions were limited. Promising supporting factors included adequately equipped and staffed urban tertiary facilities, the emergence of Stroke units, prompt assessment by health professionals, positive staff attitudes and care, two clinical care guidelines and educational and information resources being available.

**Conclusion** This review fills a gap in the literature by providing the range of opportunities and challenges to achieve health for all PWS in SA. It highlights some health system areas that show encouraging trends to improve service delivery including comprehensiveness, quality and perceptions of care.

## INTRODUCTION

Stroke is a leading cause of death and disability worldwide.[1] In South Africa (SA), stroke is the second most common cause of death

## STRENGTHS AND LIMITATIONS OF THIS STUDY

⇒ A comprehensive search strategy was developed, and the search was carried out in global, national and continental-specific databases.

⇒ The scoping review methodology included double data extraction and data review to synthesise the state of the evidence on the topic.

⇒ The use of a combination of two frameworks, the Health Systems Dynamics and Integrated People-Centred Health Services contributed to rigorous evaluation.

⇒ There was no limitation on study design or exclusion based on methodological appraisal for the inclusion of records.

⇒ Comparison of studies was challenged by heterogeneity, especially regarding design and aim.

after HIV/AIDS and a significant cause of morbidity.[2–5] It is estimated that 75 000 people experience a stroke each year in SA, contributing to 564 000 stroke-related disability-adjusted life-years.[6] Furthermore, stroke incidence in rural areas of SA is increasing; an estimated 33 500 strokes occurred in these areas in 2011, contributing to half of the national stroke burden.[7] However, these data are likely underestimated due to the absence of a national stroke database or registry and the paucity of studies that were undertaken in a few parts of the country.

Stroke is the leading cause of disability in adults in SA, placing strain on social and health services.[8] Increased prevalence of heart disease, hypertension, diabetes mellitus, behavioural factors such as smoking and structural factors such as unchecked industrialisation and urbanisation, contribute to this epidemiological transition of stroke in many low-income and middle-income countries,[9] including SA.[2] The SA government has committed to the WHO vision of achieving equitable, evidence-based rehabilitation for all by 2030.[10] SA's constitution guarantees

every citizen to have access to health services (section 27 of the Bill of Rights). The SA health system comprises the public sector (the government managed) and the private sector. Public health services are divided into primary, secondary and tertiary institutions managed by provincial Departments of Health, with the National Ministry of Health being responsible for policy development and coordination.[11] Individuals can access either public or private health services, with access to private health dependant on an individual's ability to pay for services. The majority of South Africans (84%), access health services through government-run public clinics and hospitals.[12] SA, stroke care, including rehabilitation, occurs across a range of settings, from tertiary hospitals to remote community primary healthcare facilities, and can be provided individually or in a group setting, at home, in a community environment, or a specialist centre.[2] While public health policy in SA ascribes to primary healthcare and a decentralised approach, many stroke care and rehabilitation services remain centralised at district and specialist rehabilitation hospitals.[13] It is not clear how many people access rehabilitation services following stroke, what this rehabilitation entails and how effective this rehabilitation is.[14] Therefore, achieving key global health targets and development goals will be challenging, including universal health coverage (UHC).[15 16]

UHC is achieved when every person receives essential services, such as health promotion, prevention, treatment, rehabilitation (including assistive technology) and palliative care, according to their needs and without financial hardship.[17] Accessible, responsive and quality stroke care services within a strengthened local health system will contribute to UHC for people with stroke (PWS) in SA. The extent to which UHC is currently achieved for PWS in SA is unknown.[18] We aimed to describe the health system-related factors that will facilitate UHC for PWS and the shortcomings that currently limit the implementation of UHC for stroke care in SA.

## METHODS

A scoping review was conducted according to the five-step approach recommended by Levac et al[19] as outlined in our published protocol[20]: (1) identifying the research question, (2) identifying relevant studies, (3) selecting the studies, (4) charting the data and (5) collating, summarising and reporting the results. The results are reported according to the Preferred Reporting Items for Systematic Reviews and Meta-Analyses extension for Scoping Reviews (PRISMA) guidelines.[21]

### Patient and public involvement

No patients and/or public were involved in the design, conduct, reporting or dissemination plans of this research.

### Analytical framework

This review was guided by an analytical framework adapted from the Health Systems Dynamics Framework

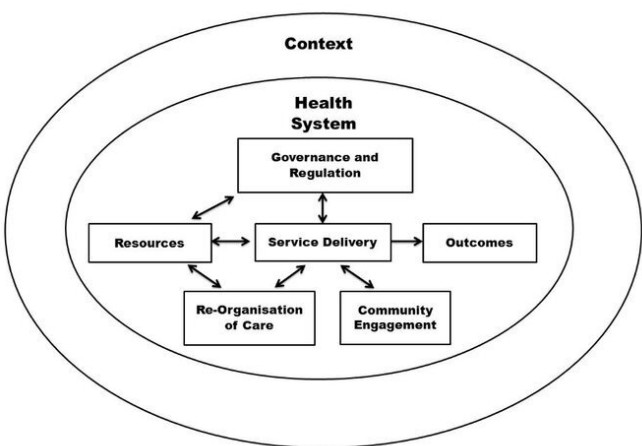

**Figure 1** Analytical framework for health system-related factors that limit or support UHC, incorporating components from the health systems dynamics framework components of the analytical framework that incorporates components from the health systems dynamics framework[22] and who framework on integrated people-centred health services and who framework on integrated people-centred health services.[23] UHC, universal health coverage.

(HSDF)[22] and WHO Framework on integrated people-centred health services (IPCHS).[23] Our analytical framework includes all the HSDF components and two components from the IPCHS: (1) Reorientation of care and (2) Enabling environment, which is appropriate to the SA context and population (figure 1). 'Resources' and 'Enabling environment' were combined and titled 'Resources' as the data items described under each were similar.

### Identifying the research question

To answer the question 'what are the opportunities and challenges within the SA health system to facilitate achieving UHC for PWS?' the review objectives were to:
1. Describe the health system-related factors that support and guide achieving universal stroke care in SA.
2. Describe the health system-related factors that limit achieving universal stroke care in SA.
3. Identify driving factors with the potential to bring change required to achieve universal stroke care in SA.

### Identifying relevant studies

In line with the purpose of scoping reviews, our approach was broad, with emphasis on studies that investigated any aspect of the healthcare system regarding stroke care in SA.

### Search strategy

We conducted a comprehensive search, according to the methodology described in our published protocol[20] and an example of the search strategy is available as a online supplemental file S1). Grey literature was identified through the National Electronic Thesis and Dissertation portal, and websites of relevant government and service

provider agencies. Field experts were contacted to identify additional relevant evidence regarding stroke care in SA. Saturation was the point at which no new records were found for inclusion.

## Eligibility criteria

Full text, SA-based studies on stroke care of any design that addressed at least one framework component were included.[20]

## Evidence selection

Two reviewers (S-MvN and KS) independently screened the titles and abstracts of identified studies. A third reviewer (I-JG) checked the results for accuracy. Results of the initial screening were compared, and full-text records were obtained for articles deemed eligible by at least one reviewer. Two reviewers (S-MvN and KS) independently screened the full texts using the eligibility criteria. Any discrepancies were resolved by discussion with a third reviewer (I-JG). Data were managed with Covidence (https://support.covidence.org/help) and Excel (V.365).

## Data charting

The six framework components were divided between three reviewers (S-MvN, KS and MYC) who extracted, collated and summarised relevant data into a purpose-built Microsoft Excel database. We considered the six components using the descriptions as outlined in online supplemental file S2 and data on the following study components were extracted:

► General study information, including author and year of publication.
► Study design, sampling and recruitment methods.
► Study settings and dates conducted.
► Population characteristics.
► Study measures.
► Research outcomes related to the framework components.

The three reviewers compared their results and reached a consensus on the organisation of extracted data. The final data and analysis were evaluated by a research team member (TS), to ensure that interpretations were credible and valid.

## Data synthesis and analysis

We summarised the study characteristics and the study designs. We used a framework analysis approach to deductively analyse data of the included studies, which consisted of five key steps as described by Ritchie et al.[24] The framework in figure 1 was used as a dynamic tool to aid this synthesis and data was managed with Atlas.ti (V.8) and Microsoft Excel (V.365).

The final synthesis of themes was confirmed following a critical discussion between all the authors. We undertook a narrative synthesis of the findings, highlighting supporting and limiting factors to achieving UHC for PWS in SA. The range of opportunities and challenges to achieve health for all PWS in SA was synthesised and included in the framework diagram.

## RESULTS

We identified a total of 4133 records and screened the abstracts of 508. After reviewing 75 full-text records, we included a total of 59 full texts in our review. A PRISMA flow diagram summarised the study selection process (figure 2).

## Study characteristics

The majority (n=41, 69.4%) of studies were conducted in urban areas, and over half of them (n=31, 52.5%) were undertaken in the Western Cape province. No studies were found from four of the nine provinces in SA (Free State, Mpumalanga, Northern Cape or the northwest provinces). The most common study design was quantitative (n=22, 37,2%), followed by mixed methods (n=14, 23.7%) and qualitative (n=10, 16.9%). Eighteen (30%) studies were community based while the remaining studies recruited participants from clinics (n=12, 20.3%) or hospitals (n=16, 27.1%). The most commonly reported framework component was Service Delivery and (n=46, 77.9%) and the least reported was Governance and Regulation (n=4, 6.7%) (table 1).

Online supplemental file S3 provides a detailed summary of included records and online supplemental file S4 provides information on components reported per included record.

Twenty-one articles (35.5%) reported on a single framework component, of which Service Delivery (n=12/21, 57.1%) was the most commonly described. The majority of articles included a combination of components (n=38, 64.4%); 24 articles (40.6%) reported on two framework components, and fourteen articles (23.7%) reported on three or more. Of the combination of components, Context was most commonly combined with Service Delivery (n=11/38, 28.9%), followed by Resources and Service Delivery (n=5/38, 13.1%).

## Service delivery

### Comprehensiveness

A comprehensive multidisciplinary team (MDT), defined as consisting of five or more different types of healthcare professionals working together in a coordinated manner, was reported in nine studies.[25–33] Two studies indicated that MDTs were either absent, limited or inefficient.[34 35]

### Continuity of care

Continuity of care was limited by poorly defined referral pathways, bed capacity for inpatient care, coordination of care and communication (among healthcare providers and with patients) in regard to care and discharge planning as well as follow-up systems. One study indicated that poor understanding of faith-based medicine by medical professionals and reciprocal lack of trust between medical and faith-based medicine practitioners

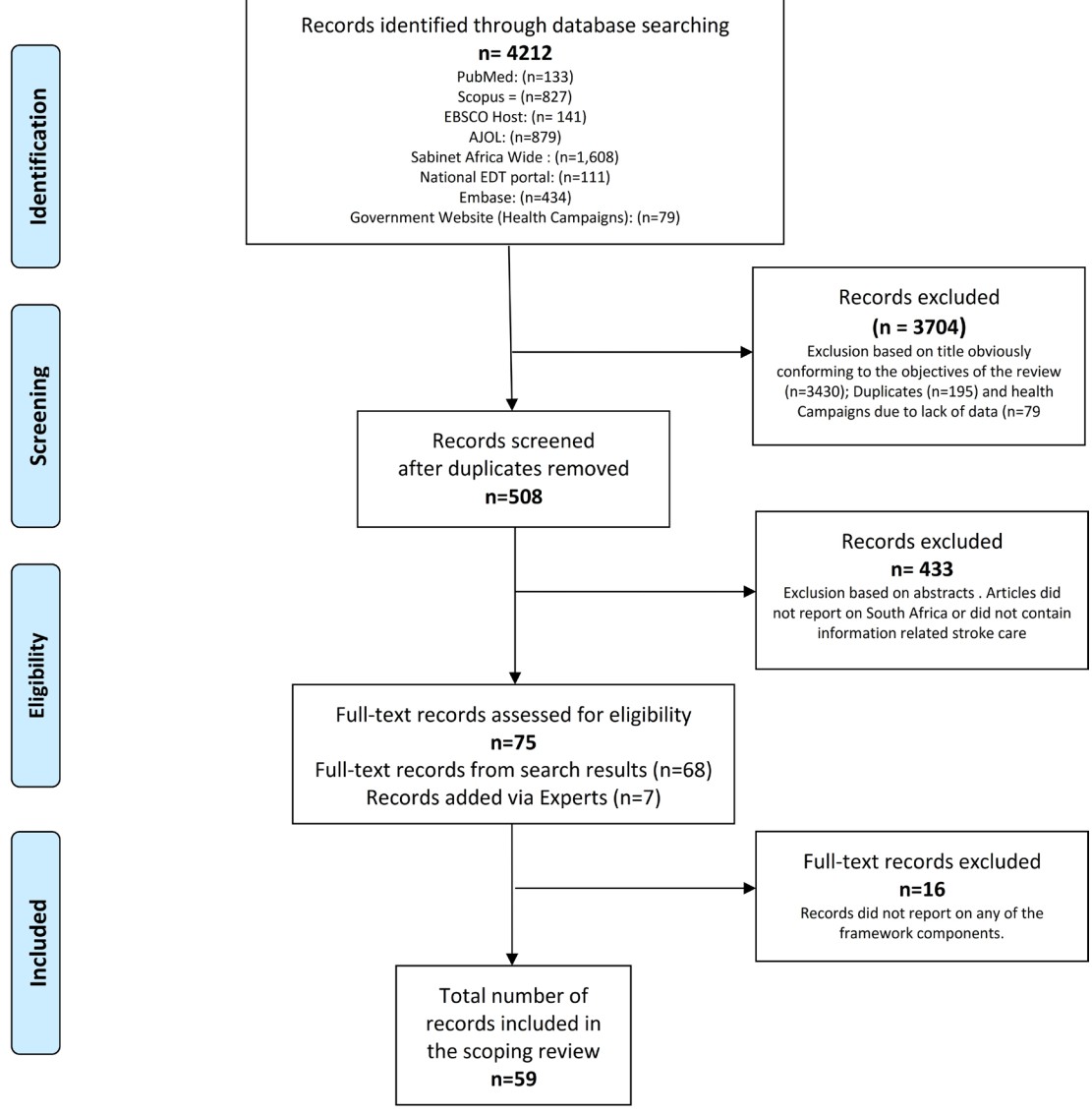

**Identification**

Records identified through database searching
**n= 4212**
PubMed: (n=133)
Scopus = (n=827)
EBSCO Host: (n= 141)
AJOL: (n=879)
Sabinet Africa Wide : (n=1,608)
National EDT portal: (n=111)
Embase: (n=434)
Government Website (Health Campaigns): (n=79)

**Screening**

Records excluded
**(n = 3704)**
Exclusion based on title obviously conforming to the objectives of the review (n=3430); Duplicates (n=195) and health Campaigns due to lack of data (n=79

Records screened after duplicates removed
**n=508**

Records excluded
**n= 433**
Exclusion based on abstracts . Articles did not report on South Africa or did not contain information related stroke care

**Eligibility**

Full-text records assessed for eligibility
**n=75**
Full-text records from search results (n=68)
Records added via Experts (n=7)

Full-text records excluded
**n=16**
Records did not report on any of the framework components.

**Included**

Total number of records included in the scoping review
**n=59**

**Figure 2** PRISMA flow chart. PRISMA, Preferred Reporting Items for Systematic Reviews and Meta-Analyses. AJOL, African Journal Online; EDT, Electronic Thesis and Dissertation.

may hinder adequate stroke care.[36] At the community level, referral to support groups lacked coordination and stroke survivors lacked knowledge of care options.[29 37–39] Two studies conducted in a rural part of the Western Cape reported that 30% (n=19) of 64 patients referred for home-based care, did not receive rehabilitation from community health workers following an assessment and treatment plan designed by a district therapist. Long waiting times contributed to a paucity of therapy sessions. Those who did receive therapy had a median of three visits that lasted 20 min each.[2 40] Waiting time for investigations such as MRI or CT scans and general stroke care was lengthy.[34 38 41] Consequently, delays in investigations were found to be associated with a significant increase in length of stay[42] and doctor-led models, where a doctor is solely responsible for the patient's care and flow of information, leading to delays in investigations and/or treatments.[40 43 44]

### Timeliness of care
Bed shortages[30 35 38 41 45] resulted in pressure to discharge patients from hospitals, which precluded rehabilitation and delayed postdischarge rehabilitation.[31 35 46 47] In addition, doctor-led models of care were reported to lead to delays in care as staff waited for instruction or referral from a doctor before conducting investigations or administering treatment.[31 37] Four studies[31 46–48] reported that patients were discharged when medically stable (average stay was 5–10 days at secondary or tertiary hospitals[30 39 46 49] despite functional deficits.[29 30 46 50 51] Cunningham[47] reviewed 168 stroke patient acute care records from the Eastern Cape province and found only 15% were referred for physiotherapy on the day of or a day before discharge.[47] Over weekends, 13% of acute care patients did not receive any therapy.[47] Difficulty with securing follow-up appointments and cancellations influenced the timeliness of postdischarge care.[2 34 40 52 53]

**Table 1** Characteristics of included records (N=59)

| Variable | Category | N (%) |
|---|---|---|
| Province | Western Cape | 31 (52.5) |
| | Gauteng | 12 (20.3) |
| | National | 6 (10.1) |
| | Eastern Cape | 4 (6.7) |
| | KwaZulu-Natal | 2 (3.3) |
| | Limpopo | 1 (1.6) |
| | Limpopo and Gauteng | 1 (1.6) |
| | Free state | 0 (0) |
| | Mpumalanga | 0 (0) |
| | Northern Cape | 0 (0) |
| | North West | 0 (0) |
| | Undefined | 2 (3.3) |
| Area | Urban | 41 (69.4) |
| | Rural and urban | 3 (5.0) |
| | Periurban | 3 (5.0) |
| | Rural | 2 (3.3) |
| | Undefined | 10 (16.9) |
| Levels of care | Community | 18 (30.5) |
| | Hospital | 16 (27.1) |
| | Primary healthcare (clinics; community health centres) | 12 (20.3) |
| | Rehabilitation centres | 6 (10.1) |
| | Undefined | 7 (11.8) |
| Study design | Quantitative | 28 (47.4) |
| | Mixed methods | 14 (23.7) |
| | Qualitative measures | 10 (16.9) |
| | Review | 2 (3.3) |
| | Editorial | 3 (5.0) |
| | Guideline | 2 (3.3) |
| Record description | Primary literature (publications) | 34 (57.6) |
| | Grey literature: dissertations | 25 (42.3) |
| Included population | PWS | 34 (57.6) |
| | Editorials and reviews | 8 (13.5) |
| | Caregiver | 6 (10.1) |
| | PWS +caregiver | 5 (8.4) |
| | PWS +HCP | 3 (5.0) |
| | HCP | 1 (1.6) |
| | PWS +HCP+experts | 1 (1.6) |
| | Traditional healers+caregivers | 1 (1.6) |
| | Policy-makers | 0 (0) |

HCP, healthcare provider; PWS, people with stroke.

## Quality of care

Four studies conducted in the Western Cape found that patients received between one and five rehabilitation sessions during acute care in hospital, except for the specialised subacute, in-patient Rehabilitation Centre where patients typically received 17 sessions.[28 31 54 55] Length of stay was typically 5–10 days and approximately 30 days in rehabilitation facilities.[42 56–59] One study reported that prompt assessment by rehabilitation professionals was associated with a shorter length of stay.[42]

## Perceptions of care

There was conflicting evidence regarding perceptions of care. Ten studies reported positive staff attitudes[32 34 40 44 59–64] while nine studies reported negative staff behaviour and attitudes.[33 36 40 52 60 65–68] A further four studies found that PWS were dissatisfied with the healthcare service along the entire continuum of care, which was driven by a lack of information about their treatment and further referral.[34 36 68] Leichtfuss[33] highlighted the significant discrepancy (p=0.00438) between doctors' understanding and patients' perception of the effectiveness of the doctors' communication; 80% (n=28) of doctors compared with 50% (n=24) of patients thought that sufficient information was communicated.[33] The study also found that patients perceived nursing services as inefficient and inadequate, which was supported by doctors who expressed the need for nursing staff who were trained in stroke care.[33] Caregiver support and training were found to be lacking[39 65 67 69] and resulted in caregiver burn-out.[66] Caregivers indicated the need for additional training and help, particularly with toileting and bath transfers, and requested more home visits by therapists.[39] Table 2 outlines measures and study findings that target Service Delivery.

## Resources

### Infrastructure

A mixed-method study by Ntamo[63] reported that substantial travelling distances were required to access rural healthcare facilities. This was echoed in Bryer's editorial on the need for community-based stroke care.[45] Makganye[60] reported that 71% of 85 rural patients (n=60) lived over 25 km away from their nearest hospital.[60] Furthermore, more specialised services often remained inaccessible[30 31 45] as their geographical location required even longer travel times. Physical access for people with a disability was further limited by poor building infrastructure (eg, no ramps, vast distances between departments) or/and uneven terrains.[70]

Three articles (longitudinal study, cross-sectional study and editorial) reported a lack of diagnostic equipment in rural facilities[26 38 45] in contrast with well-resourced urban facilities.[30 31] A mixed-methods study[63] and editorial by Taylor and Ntusi's editorial reported frequent stock-outs of basic medication at the primary care level, which resulted in additional expenses and patients' reluctance to return to rural clinics.

### Human resources

Adequately equipped urban rehabilitation centres were described in two studies.[30 31] Six studies found that high bed demand and rehabilitation workforce shortages led to high healthcare provider workloads.[30 31 34 38 45 60] Therapists reportedly treated 2–3 times more patients than

**Table 2** Supportive and limiting factors influencing different components of service delivery (N=46)

| | Service delivery | Source of evidence: Author (year) |
|---|---|---|
| **Comprehensiveness of care** | | |
| Facilitators | Comprehensive multidisciplinary teams consisting of five or more different healthcare professionals in Western Cape province | Groenewald (2017)[28]; Rhoda (2015)[27]; Joseph (2012);[29] Rouillard (2012)[26]; Leichtfuss (2009)[33] Ras (2009)[30]; Wasserman (2009)[25]; Rhoda (2009)[31]; De la Cornillère (2007)[32] |
| Barriers | Limited/absent multidisciplinary team consisting of less than five different healthcare professionals | Cawood (2012)[34]; De Villiers (2011)[35] |
| **Continuity of care*** | | |
| Barriers | Poor referral pathways (community; hospital) | Masuku (2018)[74]; Mandizvidza (2017)[38]; Cawood & Visagie (2016)[37]; Joseph (2012); De la Cornillère (2007)[32]; Kleinheibst (2007)[39] |
| | Poor follow-up and referral postdischarge | Rhoda (2014)[49]; Rouillard (2012)[26]; Bham and Ross[36]; Scheffler and Mash (2019)[2]; |
| | Lack of reciprocal respect and understanding and coordination between traditional and medical healthcare professionals | Bham and Ross[36] |
| **Timeliness of care*** | | |
| Barriers | Long queues in hospitals, community health clinics and outpatient clinics | Cawood (2012)[34]; Mudzi (2013)[53] |
| | Long waiting times for follow-up appointments | Arowoiya (2014)[52] |
| | Long waiting times for inpatients to receive specialised health services | Matshikiza (2019)[41] ; Mandizvidza (2017)[38] ; Parekh and Rhoda[51] ; Cawood (2012)[34] ; Bryer (2009)[45] |
| | Doctor-led model of care | Cawood and Visagie[40]; Cawood (2012)[34] |
| | Poor collaboration between healthcare providers | Cawood (2012)[34]; Parekh (2011) |
| | Inadequate/no rehabilitation during hospital stays | Cunningham (2012)[47]; Hilton (2011)[46]; De Villiers (2009)[48]; Rhoda (2009)[31] |
| **Quality of care** | | |
| Facilitators | Prompt assessment by an allied health professional significantly decreases the length of stay | Viljoen (2014)[42] |
| Barriers | Lack of appropriate care due to lack of stroke-specific knowledge | Mandizvidza (2017)[38]; Leichfust (2009); Ras (2009)[30] |
| | Insufficient no of in-patient rehabilitation sessions | Groenewald and Rhoda[28]; Parekh (2011); Rhoda et al (2011)[55]; Rhoda (2009)[31] |
| | Short length of stay at all levels of care except for specialist rehabilitation facilities | Groenewald (2018)[59]; Mabunda (2015)[58]; Rhoda (2014)[49]; Viljoen (2014)[42]; Hilton (2011)[46]; Parekh (2011); Blackwell and Littlejohn[56]; Mudzi (2010)[50]; Ras (2009)[30]; Kleinhebst (2007); Felemengas (2004)[57] |
| **Perceptions of care** | | |
| Facilitators | Positive staff attitudes and care | Taylor and Ntusi[64]; Groenewald (2018)[59]; Kotsokoane (2018)[61]; Hossain (2016); Kusambiza-Kiingi (2016)[62] ; Cawood and Visagie[40]; Bham and Ross[36]; Cawood (2012)[34]; Ntamo (2011)[63]; De la Cornillère (2007)[32] |
| Barriers | Negative staff attitudes and behaviour for example, impersonal care; inappropriate support; poor communication; lack of cultural sensitivity, rudeness and delayed assistance with patient's personal hygiene | Smith (2019)[68]; Cawood and Visagie[40]; Makganye (2015)[60]; Posner (2015)[67]; Arowoiya (2014)[52]; Leichtfuss (2009)[33]; Thomas and Greenop[66]; Bham and Ross[36]; Biggs (2005)[65] |
| | Dissatisfaction with healthcare received | Arowoiya (2014)[52]; Cawood (2012)[34]; Bham and Ross[36] Ntamo (2011)[63]; Kleineibst (2007)[39] |
| | Lack of caregiver training | Kusambiza-Kiingi (2017)[72]; Mashau et al (2016)[69]; Mudzi (2010)[50]; Kleineibst (2007)[39]; Rouilliard (2012); Felemengas (2004)[57] |

*No supporting factors reported.

the daily recommendation.[30] Mandizvidza[38] reported that nursing shortage at all healthcare levels in rural KwaZulu Natal negatively impacted basic stroke care. However, better-resourced urban tertiary hospitals in the Western Cape were also reported to experience staff shortages.[38]

A quantitative cross-sectional study found that rehabilitation services are severely limited at the primary care level with half of the community health centres in the Western Cape providing rehabilitation services, and only two offering speech therapy.[31] Stroke care was often provided

by healthcare professionals without specific stroke-related training[30 33 38] (table 3).

None of the included articles reported on financial allocations for stroke care.

## Context

### Well-being and caregiver factors

Two longitudinal studies and one retrospective survey reported mental health problems such as anxiety and depression among PWS and caregivers.[26 27 71] PWS also described feelings that related to confinement, personality changes, imposed family adjustments and caregiving burden.[50 57 72] Gender bias in caregiving roles was reported where women commonly left employment to assume caregiving responsibilities of male partners or parents[46] or children cared for women with stroke.[47 57 70]

### Financial implications

Financial burden was found to increase when spouses became primary caregivers (without gainful employment) or through the employment of additional caregivers.[57] Costs poststroke were high due to additional caregiving expenses[60 73] and studies found that there was limited access to disability-support, old age-support or child-support grants.[52 65] The financial burden among rural stroke survivors was compounded by low income before the stroke, difficulty in obtaining social grants due to limited awareness of eligibility criteria and the application processes, and lack of transport to submit grant applications.[53 66] Poverty impacted access and utilisation of rehabilitation as available finances were preferentially used to meet basic needs such as food.[74]

### Access to transport

Six studies reported transport being a limiting factor to access care due to expensive private transport, unreliable public transport, and inflated costs of a trip to accommodate assistive devices.[32 37 39 63 65 75]

### Cultural beliefs and health literacy

Two qualitative case studies reported that PWS in SA held cultural beliefs regarding the cause and recovery of strokes, such as ascribing stroke to witchcraft or religious beliefs.[36 60] Poor health literacy[60 66 68] and these beliefs further affected the care-seeking ability of communities. Bham and Ross[36] reported that healthcare professionals needed greater awareness of cultural practices, such as the inclusion of extended family in decision-making procedures, adaption of communication style when interviewing older persons, and sensitivity to religious and traditional beliefs, to facilitate the inclusion and full participation of marginalised communities.[36]

## Community engagement

### Self-efficacy

Leichtfuss[33] found that PWS and/or their caregivers believed that they were not involved in decision making with regard their care. Felemengas[57] and Cawood[34] reported that PWS were neither confident with self-health

**Table 3** Facilitators and barriers influencing different components of resources (n=16)

| | Resources | Source of evidence: Author (year) |
|---|---|---|
| **Infrastructure** | | |
| Facilitators | Adequate equipment (urban rehabilitation centre setting) | Ras (2009)[30]; Rhoda (2009)[31] |
| Barriers | Lack of equipment (rural setting) | Mandizvidza (2017)[38], Cawood (2012)[34]; Cunningham (2012)[47]; Rhoda (2009)[31] |
| | Inadequate no of ambulances; ineffective systems to request an ambulance | Mandizvidza (2017)[38], Biggs (2005)[65] |
| | Poor accessibility of health centres due to location, building structure, or terrain surrounding the health facility | Maleka (2012)[70], Ntamo (2011)[63], Bryer (2009)[45], Rhoda (2009)[31] |
| | Insufficient no of beds or hospitals due to fiscal problems | Matshikiza (2019)[41]; Mandizvidza (2017)[38]; De Villiers (2011)[35], Bryer (2009)[45], Ras (2009)[30] |
| | Inadequate special investigations and infrastructure for diagnosis and management | Mandizvidza (2017)[38], Viljoen (2014)[42] (2016); Bryer (2009)[45] |
| | Frequent medication outages | Taylor and Ntusi[64], Ntamo (2011)[63] |
| **Human resources\*** | | |
| Barriers | Staff shortages | Mandizvidza (2017)[38], Makganye (2015)[60]; Cawood (2012)[34], Bryer (2009)[45], Ras (2009)[30]; Connor (2005)[89] |
| | Lack of stroke-care specific training for staff | Mandizvidza (2017)[38], Leichfust (2009); Ras (2009)[30], Kleineibst (2007)[39] |

\*No supporting factors reported.

management nor satisfied with predischarge training and information.[34 57] A large mixed-methods study[65] that included a survey (N=418) reported that PWS and caregivers lacked awareness of the availability and benefit of rehabilitation services or support groups and this was echoed by Burton's editorial.[76] Cawood *et al*[13] found that nearly half (n=53; 47%) of the participants in their cross-sectional study indicated via a survey that they did not receive assistance from stroke organisations.[40] Low participation in a peer support programme was found[29] despite patients who attended stroke support reporting better self-efficacy and feeling supported.[34 65]

## Community integration

PWS were not fully reintegrated into their communities[61 77] due to negative attitudes of family, friends and society.[34] Inaccessible community activities (28.3%), poor mental health (18.9%),[78] financial constraints (45.3%)[77] and inaccessible transport[65] contributed to limited community integration. Fear of stigmatisation,[70] functional dependency especially due to incontinence,[32 37 50 63 79] and fear of becoming a victim of crime[40] also limited integration.

## Homecare resources

A Stroke Home Care booklet (in different languages) was developed for the SA context.[80] In focus group discussions, seven-stroke survivors (n=15; 46%) demonstrated improved knowledge, confidence and ability to communicate information about their stroke after using the booklet.[80] However, the sample included in this study was small and the booklet was only available in English when acceptability was tested. The stories and pictures were found to be culturally sensitive.[80]

## Reorganisation of care
### Educational and information resources

Two educational resources were available via institutional websites for the public: The stroke Home Care booklet[80] and the SA contextualised Bridges Stroke Self-management intervention workbook.[59] The MyStroke website (www.mystroke.co.za) was developed following a public health awareness campaign and lists available stroke care centres and services for better coordination.[76] The mySOS app is an e-health initiative that directs and connects users with emergency care, potentially improving the timeliness of care. In rural settings, telemedicine was used to connect with specialist services.[81] However, none of these resources included efficacy trials or determined the usage of the website or application.

## Stroke unit

At a central hospital in Western Cape, the stroke unit was associated with reduced mortality and increased rehabilitation referral, staff training and family involvement in treatment decisions.[48] Stroke units were recommended in evidence-based SA stroke care guidelines.[13 82]

## Palliative care integration

Findings based on focus groups of patients recommended that palliative care should be incorporated into stroke care. However, better education of all stakeholders on palliative care benefits was needed.[44]

## Governance and regulation

Two-stroke clinical care guidelines for SA were identified.[13 82] One focused on acute and postacute stroke care,[82] and the other on stroke rehabilitation.[13] Mandizvizda[38] evaluated the level of adherence to the acute stroke care guideline in all levels of care in the Western Cape province and reported poor adherence in primary, secondary, and tertiary hospitals (general wards), with the two Stroke Units (situated in tertiary hospitals), being the most compliant.[38] Challenges to adherence of the guidelines included staff shortages, limited access to diagnostic investigations, and delays in patients presenting to healthcare facilities.[38]

There were no national stroke-specific policies. While many people with disabilities are reliant on financial support from the government through grants, there was no specific policy on financial support for PWS or their caregivers. Poor intersectoral coordination between government departments was found with regard the responsibility for policy concerning persons with disabilities.[83] Governance and Regulations was the most limited component reported, which demonstrates a deficit in leadership and policy for how stroke care should be implemented and conducted at all levels of care.

## Limiting and supporting factors

Health system limitations and factors that support the achievement of UHC for PWS in SA are presented in figure 3. Findings of each health system component of the framework are mapped and identify challenges and opportunities that speak to stroke care in the public sector.

## DISCUSSION

This scoping review summarises the available evidence of achieving health for all PWS in SA. Included articles evaluated a diverse range of health system categories and various outcomes, with the majority of studies reporting on two or more framework components. There were several key limiting factors towards achieving UHC, which included a lack of governmental regulation in terms of stroke policies and guidelines poor timeliness of care, a lack of the continuity of care and a lack of a comprehensive MDT at rural health facilities. Furthermore, bed and staff shortages and a lack of stroke-specific training, poor access to acute care and diagnostic equipment contributed to limiting UHC. Regular medication stockouts, lack of caregiver training and negative perceptions of care were also found to be important limiting factors. There were also many supporting factors toward achieving UHC for PWS in SA, which included adequately equipped and

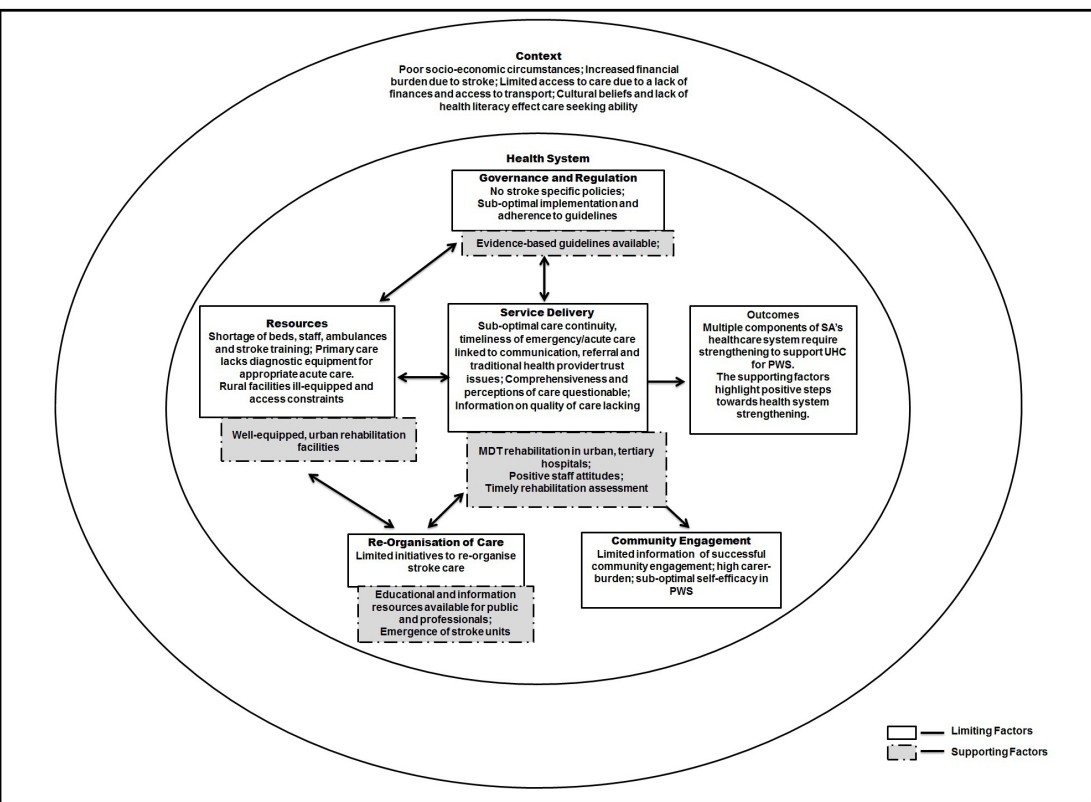

**Figure 3** Limiting and supporting factors towards achieving UHC. MDT, multidisciplinary team; PWS, people with stroke; SA, South Africa;UHC, universal health coverage.

staffed urban tertiary facilities, the emergence of Stroke Units in urban areas, prompt assessment by rehabilitation professionals, and positive staff attitudes and care. Resources that were available to support achieving UHC include two clinical care guidelines, and educational and information resources being available online. Drivers to achieve UHC for PWS in SA may include better governance and regulation to mitigate fiscal shortfall that has resulted in infrastructure and human resource limiting factors, intersectoral collaboration between government departments to assist with access to social support, and reliable and affordable transport to access healthcare.

### Limiting factors

A key finding of this review is a lack of adequate Governance and Regulations in terms of government support and investment in policy and treatment guidelines, resource distribution and commitment to evidence-based solutions (eg, stroke units). Equity for people with disabilities, including PWS, requires a concerted commitment from the SA government to ensure that UHC for all is achieved.[84] Opportunities to facilitate these renewed efforts include administrative interventions by both government and hospital management to address system-based limiting factors, such as access to patients' medical records and obtaining appointments. Addressing staff shortages and improving stroke-specific training may mitigate the excessive workload of healthcare workers and improve service delivery, as was achieved during a

pilot programme in Namibia, where an increase in the number of nurses resulted in improved service delivery.[85] However, attracting and retaining health professionals in rural and remote areas is multi-factorial[86 87] and contextual strategies to attract and retain health professionals in these areas are needed.[86] Dedicated stroke units in hospitals have reduced stroke mortality, increased access to rehabilitation from MDTs, and have resulted in improved discharge planning services at these stroke units compared with managing PWS in general medical wards.[48] Political and leadership support for these units may contribute to better stroke outcomes and improve community reintegration and return to work for PWS in SA.

Service Delivery and Context-related factors were most frequently reported in combination and were consistently reported as poor. Findings of poor timeliness of care, a lack of continuity of care, and absence of comprehensive MDTs in rural areas are similar to health system weaknesses found in Rwanda and Malawi.[88] The main hindrances affecting service delivery in SA related to training, resources and communication channels. Poor referral networks and few rural rehabilitation facilities were compounded by inadequate caregiver training, lack of stroke-specific staff training, bed shortages and diagnostic equipment. As a result, many PWS are lost to follow-up care leading to poor management of comorbidities and potentially placing patients at risk of recurrence

and secondary complications such as spasticity, pressure sores, aspiration pneumonia and mobility difficulties.[89 90]

Access to equitable and affordable healthcare for PWS may be affected by contextual factors outside the healthcare system. Social determinants of health (poverty, education) and general safety can be addressed through intersectoral collaboration. Social service and health sector collaboration may ensure that eligible PWS are aware and have access to social grant support. This was echoed as an international need in a scoping review which included studies from North America, the UK and Europe.[88] Cooperation between both private and public transport services, and the health sector is needed to find a solution for accessible, affordable and reliable transport for PWS and their caregivers. While there is strong evidence of the link between lack of access to transport and negative effects on healthcare, research on possible solutions and the effectiveness thereof is scarce.

### Supporting factors

Despite the many limiting factors that were described, there are promising supporting factors to achieve UHC for PWS in SA. Well-equipped rehabilitation facilities in urban areas, comprehensive MDTs in urban, tertiary hospitals and a stroke unit in an urban area are already in place. There were also two clinical care guidelines and educational and information resources were available. Although some PWS reported their dissatisfaction with the care they received several studies reported patient and caregiver satisfaction, as well as positive staff attitudes, which were perceived to facilitate physical improvement through rehabilitation compliance. This was consistent with findings where the attitude and emotional approach of health professionals, as well as caregivers, affected the level of motivation for rehabilitation attendance in PWS in an inner city teaching hospital in a high income setting.[91] Maclean *et al*[91] found that a positive rapport between patients and healthcare providers resulted in increased motivation and easy transmission of information about rehabilitation.

### Implications for future research

The limited supporting factors and a multitude of limiting factors reported in the included articles of this scoping review highlight the gaps that remain and present opportunities for future research. Key questions include the effect of continuity and timeliness of care, and perceptions of care on the improvement of service delivery, as well as the effect of resources (such as staffing, bed allocation and access to diagnostic equipment) and the impact of stroke-related training on service delivery. The distribution of research as reported in this review was found to be disproportionate with just over half of the studies being conducted in a single province (Western Cape) and largely in urban areas, with four of the nine provinces not being reported on at all. Insights regarding barriers and facilitators to UHC for PWS residing in these unreported provinces are warranted.

Future research may focus on:

► Strategies to coordinate care for multimorbidity (eg, combined appointments with different health professionals) to minimise financial hardship on healthcare users and to evaluate effective and efficient holistic management of health, compared with silo treatment approaches.
► Extension of research on stroke services in the under-reported provinces.
► Evaluation of accessible, quality services beyond urban areas.
► Development and testing of stroke-specific capacity development for staff that is evidence based, patient centred and holistic. Factors to highlight in training may include cultural responsiveness and awareness of the social determinants of health.
► Strategies to improve and implement person-centred discharge planning, which should include caregiver training and support before and after discharge.
► Development and evaluation of sustainable strategies to provide peer support groups either in person or on a digital platform, for both PWS and their caregivers, to provide ongoing support.
► Innovative public health campaigns via social media, television, or radio to increase the awareness of stroke signs and the urgency of seeking help. The impact, reach and process evaluation of such campaigns should monitor effectiveness.

### Strengths and limitations of the scoping review

We used a comprehensive search strategy that followed PRISMA guidelines, and robust methods that included double data extraction and review to produce a comprehensive state of the evidence. Our framework for analysis included a people-centred framework that acknowledged that health service provision should be coordinated around people's needs and preferences and provided in a way that is safe, effective, timely, affordable and of acceptable quality. The framework also acknowledged the political context and the social and economic determinants of health. However, this review has limitations. The disproportioned distribution of where research on stroke care services was conducted may have limited generalisability. We included research articles, dissertations and commentaries, and may have missed evidence indexed in health or government websites.

### CONCLUSION

Stroke is the leading cause of disability in adults in SA, which places strain on national social and healthcare services and the SA government has committed to the WHO vision of achieving equitable, evidence-based healthcare for all by 2030. However, his review highlights health system components such as Governance, that requires strengthening, to enhance readiness for UHC for PWS.

Despite the available guidance on the best strategies to support healthcare systems in delivering stroke care services, the main findings of this review show that the stroke care services for PWS in SA are limited with a strong urban bias. The findings of this review have highlighted health systems challenges that speak to inequitable stroke care in the public sector. Health system strengthening driven by good governance and regulation of health services, continuity and timeliness of care, accessible facilities and well-equipped rehabilitation services is urgently needed. Health system limitations are compounded by contextual factors, highlighting the need for health system strengthening strategies that are tailored for the local context.

This scoping review highlights some health system areas that show encouraging trends to improve service delivery including comprehensiveness, quality and perceptions of care. The results of this review can be used to inform policymakers and healthcare professionals of healthcare system challenges and opportunities to effectively move towards UHC for PWS in SA. Governments should be held accountable for stroke care in terms of financial resource allocation, and prioritise this marginalised group in the proposed national health insurance scheme.

**Author affiliations**
¹Division of Physiotherapy, Department of Health and Rehabilitation Sciences, Stellenbosch University, Stellenbosch, Western Cape, South Africa
²SACDIR Indian Institute of Public Health Hyderabad, Public Health Foundation of India, New Delhi, India
³International Center for Evidence in Disability, Clinical Research Department, London School of Hygiene & Tropical Medicine, London, UK
⁴Rehabilitation Sciences, Stellenbosch University Faculty of Medicine and Health Sciences, Cape Town, Western Cape, South Africa
⁵Department of Global Health and Development, Faculty of Public Health and Policy, London School of Hygiene and Tropical Medicine, London, UK
⁶Department of Disease Control, Faculty of Infectious and Tropical Diseases, London School of Tropical Health and Medicine, London, UK
⁷Global Health, Stellenbosch University Faculty of Medicine and Health Sciences, Cape Town, Western Cape, South Africa
⁸Clinical Research, London School of Hygiene and Tropical Medicine, London, UK

**Contributors** S-MvN and KS in consultation with all authors constructed the search. S-MvN, KS and MYC extracted all data in consultation with all authors. S-MvN, KS, I-JG, JW, QAL and TS analysed the extracted data. S-MvN drafted and revised the paper. KS, I-JG, MYC, SF, RE, JW, QAL and TS reviewed the manuscript and provided feedback on the drafts. All authors read and approved the final manuscript. TS was the guarantor for this work.

**Funding** This research was commissioned by the National Institute for Health Research (NIHR) Global Health Policy and Systems Research Development Award using UK aid from the UK government. Grant number NIHR130180.

**Disclaimer** The views expressed in this publication are those of the author(s) and not necessarily those of the NIHR or the Department of Health and Social Care.

**Competing interests** None declared.

**Patient consent for publication** Not applicable.

**Provenance and peer review** Not commissioned; externally peer reviewed.

**Data availability statement** All data relevant to the study are included in the article or uploaded as online supplemental information. The protocol for this scoping review was published in BMJ Open. http://dx.doi.org/10.1136/bmjopen-2020-041221.

**ORCID iDs**
Sureshkumar Kamalakannan http://orcid.org/0000-0003-4407-7838
Maria Yvonne Charumbira http://orcid.org/0000-0002-2441-2566
Tracey Smythe http://orcid.org/0000-0003-3408-7362

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
