## [Reviewer comments · BMJ Open]

ARTICLE DETAILS

TITLE (PROVISIONAL)	Towards universal health coverage for people with stroke in South Africa: a scoping review
AUTHORS	Sjan-Mari, van Niekerk; Sureshkumar, K; Gakeemah, Inglis-Jassiem; Charumbira, Maria; Fernandes, Silke; Webster, Jayne; English, Rene; Louw, QA; Smythe, Tracey

VERSION 1 – REVIEW

REVIEWER	Osborne, Candice The University of Texas Southwestern Medical Center
REVIEW RETURNED	06-Jul-2021

GENERAL COMMENTS	Thank you for the opportunity to review this manuscript. It is well written, though there are concerns that need to be addressed. -Spell out mnemonics in abstract and with initial use in the paper.-The authors assume the readers are knowledgeable of the healthcare system in SA and the typical healthcare pathway of patients with stroke. It would help if the authors provided a brief overview of the SA healthcare system (socialized?, privatized?, etc).-Can the authors provide the search terms that were used to gather the studies included.-The quality of each study has not been appraised. Please provide the level of evidence for each study. The manuscript may offer valuable information, but it is impossible to determine how sound the results of this scoping review are without an understanding of the overall level of rigor of the studies included.-Results-general comments-add that the majority of the studies included take place in an urban setting in the characteristics section.-when describing the results of specific studies, it would be helpful to include the study setting. It is difficult to decipher from where along the healthcare continuum the data is being gathered. For example—under the Timeliness of Care section—line 48—168 records were reviewed and only 15% received referrals for physio prior to d/c. D/c from where? Acute care? Line 49: over weekends, 13% of patients did not receive therapy—in acute care? at home?
---

	-The results under the service delivery section are unclear. -Is 'comprehensiveness' defined as a MDT consisting of 5 or more health care professionals? This is confusing. -'continuity of care'—this section is confusing. What is 'poor referral'? What do you mean by 'coordination'? communication—provider to provider or provider to patient or both? Communication about what...care plan? Discharge? Referrals? -What do you mean by traditional medicine? Poor understanding by the patient or the provider? -lack of trust between whom? -30-40% of patients did not receive home-based care—why? Was it warranted? Is home-based care the norm in SA? -Waiting time for investigations—please define 'investigation'. What is this? -What is a doctor-centric model? -Timeliness of care -What does the pressure to d/c patients stem from...lack of beds, insurance payments? Is this d/c from acute care, subacute care, inpatient rehabilitation? -review of 168 patient records...Acute care records? inpatient rehabilitation records? skilled nursing facility records? -Quality of care - 5 rehab sessions where? In acute care? -what is a specialized rehab centre—is this inpatient? -Perceptions of care -positive and negative attitudes about what? -Dissatisfied with healthcare services where...in the hospital? In general? The whole continuum? -Line 18 is confusing. Doctors' understanding of what? -Resources -Human resources -line 51: negatively impacted? -Context -Cultural beliefs and health literacy
--	---

	-line 29: care seek ability of communities...meaning communities of people with these beliefs combined with poor health literacy? -Discussion - page 19 paragraph about reported supporting factors: Here the authors list many supporting factors that were not included in the Results section. This suggests a biased presentation of the results and calls into question the integrity of this scoping review. The authors' first objective was to describe factors that support and guide achieving universal stroke care in SA, yet the supports are not included in the Results section. Also, Figure 3: Limiting and supporting factors towards achieving UHC, only includes the limiting factors. A scoping review should present ALL of the evidence.
--	--

REVIEWER	Katzenellenbogen, Judith University of Western Australia, School of Population and Global Health
REVIEW RETURNED	18-Jul-2021

GENERAL COMMENTS	Towards universal health coverage for people with stroke in South Africa: a scoping review Thank you for the opportunity to review this scoping review investigating opportunities and challenges to achieve University Health Coverage for people with stroke in the South African health system. This manuscript follows the protocol paper published in BMJOpen in 2020. The authors can be congratulated on synthesising the various studies into a meaningful whole. Abstract: Please define UHC, PWS, WHO when first used in the abstract. This may be all people read. Introduction: Please define all abbreviations when first used in the paper (UHC, PWS, SA) Given that this is an international journal, you need to provide a short paragraph on the health system in South Africa in the context of a federal political structure and who is responsible for the services you are covering in your review. Fig 1: Please provide a more meaningful title that makes it clear to a reader who may not cover all the content in the text. What is the framework of/ for? Also acknowledge the two frameworks that this one draws on:
--

Eg Source: this framework incorporates components from the XXX framework (ref) and the yyyy framework (ref)

Search strategy: I know you have stated this in the protocol paper, but I would have liked to see the terms you searched on. You also have not indicated the years that you covered. It is annoying to have to go to the protocol paper for such basic information.

You mention that you have searched to government websites; I would have liked some information about the number, distribution and service components of stroke units and, if possible, health professionals involved in stroke services around the country. This is part of the broader stroke context. Given that you did not go beyond accessing research papers/dissertations and opinions/commentary, you should mention this as a limitation of your review.

Figure 2 is unreadable.

Table 1: this is not very informative and is unidimensional. The framework components do not fit into the layout of the table as a whole. I suggest that you create a matrix (this will need to be in landscape), with the framework components each having a column and the other variables each having their rows. You can consolidate the 4 provinces with no records into one. Change 'Area' to 'Area type'

Variable

CE

Con

ReO

Res

SD

Total (%)?

Province

Western Cape

Gauteng

etc

Area type

Urban

Rural/urban

I am not sure of what your crosses mean, but think you can put actual numbers of papers covering that component in each variable category. eg It could be that in WC the studies cover multiple components so that you can't total them in a row. So make the total for the column. Think about how this table can help readers get an overview of the papers and what they covered.

Please help the reader by putting the legend explaining the abbreviations in the same order as the components in the table.

Please explain what you mean by 'doctor-centric'. It can mean different things to different people. Do you mean overemphasis on biomedical or just to suit the doctors' schedules or what? And explain why is this a problem?

pg 11 Line18. Sentence not understandable – rephrase.

Table 2: rephrase title?

Supportive and limiting factors influencing different components of service delivery

Put the service delivery column on the left and the papers from which you derived the evidence on the right. This new column on the right should have a column heading 'Source of evidence: Author (year)
Write out MDT in full.

Pg 14, Line 3: information systems are part of infrastructure, not human resources.

Should there be a section or more mention of financial resourcing? Even if in the intro under health system structure and funding.?

Table 3: similar changes to Table 3 in terms of layout and title.

Discussion:
Pg 21; line 16.

Start limitations as new paragraph. Or else change the sentence to improve flow.

Replace 'There was no limitation on study design...' to 'There was no restriction on study design...'

Pg 21, line 30: no need to redefine WHO if you have done so previously in the paper.

	Is there room somewhere for a statement on primary prevention of stroke (I accept that care post-stroke is covered by your paper)?
--	---

VERSION 1 – AUTHOR RESPONSE

Reviewer #1:

1. Spell out mnemonics in abstract and with initial use in the paper.

We have spelled our mnemonics in the abstract and initial use in the paper as suggested.

2. The authors assume the readers are knowledgeable of the healthcare system in SA and the typical healthcare pathway of patients with stroke. It would help if the authors provided a brief overview of the SA healthcare system (socialized?, privatized?, etc).

We have included a brief overview of the South African healthcare system in the introduction:

South Africa’s Constitution guarantees every citizen to have access to health services (section 27 of the Bill of Rights). The SA health system comprises the public sector (government managed) and the private sector. Public health services operate at primary (community), secondary and tertiary levels of healthcare. The public health sector’s policies are governed by the National Ministry of Health while implementation of healthcare is managed by Provincial ministries of Health (11). South African citizens have access to either public or private health services, depending on preference, the ability to pay for services and private health insurance. The majority of South Africans (84%), access health services through government-run public clinics and hospitals because they cannot afford private medical care or insurance (12). In SA, stroke care occurs across a range of settings, from tertiary hospitals to remote community primary healthcare facilities, and can be provided individually or in a group setting, at home, in a community environment or a specialist centre (2). Whilst public health policy in SA ascribes to primary health care and a decentralised approach, provision of stroke services remains centralised at district and specialist rehabilitation hospitals (13).

3. Can the authors provide the search terms that were used to gather the studies included.

We have included our search terms in Supplementary file 1 and added a note in our methods section on page 6

‘and an example of the search strategy is available as supplementary file (S1).’

4. The quality of each study has not been appraised. Please provide the level of evidence for each study. The manuscript may offer valuable information, but it is impossible to determine how sound the results of this scoping review are without an understanding of the overall level of rigor of the studies included.

We acknowledge that a scoping review is a process of mapping the existing literature based on a broad topic (unlike a systematic review which is typically underpinned by a narrow review question). Scoping reviews can be also be used to identify research gaps across multiple, complex elements/related research fields. Scoping reviews can also inform systematic reviews e.g. to identify appropriate parameters of a review (i.e. define the targeted population, intervention, comparison, outcomes).

Due to their broad nature, scoping reviews do not formally evaluate the quality of evidence (unlike systematic reviews) and often gather information from a wide range of study designs and methods. The PRISMA extension for Scoping reviews, which is currently the globally accepted reporting guideline for scoping reviews (<https://www.acpjournals.org/doi/full/10.7326/M18-0850>) states that appraisal is not expected for scoping reviews and as well as the following statement pertaining to critical appraisal “If done, provide a rationale for conducting a critical appraisal of included sources of

evidence; describe the methods used and how this information was used in any data synthesis (if appropriate).” We are therefore unable to justify the usefulness of a critical appraisal in our scoping review as the heterogeneity of the study design prohibit any standardised approach to synthesize the evidence, and the broad nature of the scoping review questions do not lend themselves to specific type of evidence.

Armstrong R, Hall BJ, Doyle J, Waters E. Cochrane Update. 'Scoping the scope' of a cochrane review. *J Public Health (Oxf)*. 2011 Mar;33(1):147-50. doi: 10.1093/pubmed/fdr015. PMID: 21345890.

Results. General comments:

5. add that the majority of the studies included take place in an urban setting in the characteristics section.

We have included on page 8 of the characteristics section: the majority (69.4%) of studies were conducted in urban areas

6. when describing the results of specific studies, it would be helpful to include the study setting. It is difficult to decipher from where along the healthcare continuum the data is being gathered. For example—under the Timeliness of Care section—line 148—168 records were reviewed and only 15% received referrals for physio prior to d/c. D/c from where? Acute care? Line 49: over weekends, 13% of patients did not receive therapy— in acute care? at home?

Thank you for your suggestion of clarifying where along the healthcare continuum the data is being gathered. We have revised the text as follows in Page 10-11: “Cunningham (2012) (47) reviewed 168 stroke patient acute care records from the Eastern Cape province and found only 15% were referred for physiotherapy on the day or day prior to discharge from in-patient acute care (47). Over weekends, 13% of acute-care patients did not receive any therapy (47)”

In addition, we have included further clarification in pages 10, 11

“Three studies conducted in the Western Cape found that patients received between one and five rehabilitation sessions during acute care in hospital, except for the specialised sub-acute, in-patient Rehabilitation Centre where patients typically received 17 session”

” Waiting-time for investigations such as magnetic resonance imaging or computerised tomography-scans and care was lengthy (34,38,41).”

“One study reported that prompt assessment by rehabilitation professionals during acute care was associated with shorter length of stay (42).”

“five rehabilitation sessions during acute care in hospital...”

“specialised sub-acute, in-patient Rehabilitation Centre”

“dissatisfied with the healthcare service along the entire continuum of care, which was”

7.The results under the service delivery section are unclear. -Is ‘comprehensiveness’ defined as a MDT consisting of 5 or more health care professionals? This is confusing.

Our apologies for the confusion, the sentence was revised for clarity in Page 10: “A comprehensive multi-disciplinary team (MDT), defined as consisting of five or more different types of health care professionals working together in a coordinate manner, were reported in nine studies (25-33).”

8. ‘continuity of care’—this section is confusing. What is ‘poor referral’? What do you mean by ‘coordination’? communication—provider to provider or provider to patient or both? Communication about what...care plan? Discharge? Referrals?

Thank you for suggesting clarification. We have reworded the sentences as follows in page 10: “Continuity of care was limited by poorly defined referral pathways, bed capacity for inpatient care, coordination of care, communication (among healthcare providers and with patients) in regard to care and discharge planning as well as follow-up systems”

Coordination of care is defined as “The deliberate organization of patient care activities between two or more participants (including the patient) involved in a patient’s care to facilitate the appropriate delivery of healthcare services.” McDonald, K. M., Sundaram, V., Bravata, D. M., Lewis, R., Lin, N., Kraft, S. A., ... & Owens, D. K. (2007). *Closing the quality gap: a critical analysis of quality improvement strategies* (Vol. 7: Care Coordination).

9. What do you mean by traditional medicine?

In this specific context traditional medicine refers to a form of health medicine practiced by the Imam in the Muslim faith. We have changed the term traditional medicine to faith-based medicine (see below) to clarify.

Poor understanding by the patient or the provider? -lack of trust between whom?

The sentence was revised and reads as follows on page 10: "One study indicated that poor understanding of faith-based medicine by medical professionals and reciprocal lack of trust between medical and faith-based medicine practitioners may hinder adequate stroke care (36)."

10. 30-40% of patients did not receive home-based care—why? Was it warranted? Is home-based care the norm in SA?

Although home-based care is not the norm in most of South Africa, but in the Western Cape where the two studies were conducted, home-based care services delivered by community health workers were available. A district-based therapists performs the assessments and designs treatment plans, which are executed by community health workers no specific rehabilitation training. We have revised the text as follows on page 10:

"Two studies conducted in a rural part of the Western Cape reported that 30% (n=19) of the 64 patients who were referred for home-based care, did not receive rehabilitation care as delivered by community health workers following an assessment and subsequent treatment plan designed by a district therapist. The lack of therapy sessions was due to a long waiting time for appointments. Those who did receive therapy, had a median of three visits which lasted 20 minutes each (2, 40)."

11. Waiting time for investigations-please define 'investigation'. What is this?

The following sentence was added to clarify 'investigation' on page 10: "Waiting-time for investigations such as magnetic resonance imaging or computerised tomography-scans and care was lengthy (34,38,41)."

12. What is a doctor-centric model?

We have included the following to clarify the meaning of a doctor-centric model on page 10:

"Findings included delays in investigations being associated with a significant increase in length of stay (42) and doctor-led models, where a doctor is solely responsible for the patient's care and flow of information, delaying investigations or treatments (40,43,44)"

"In addition, doctor-led models of care were reported to lead to delays as staff wait for instruction or referral from a doctor before conducting investigations or administering treatment (31, 37).

13. Timeliness of care -What does the pressure to d/c patients stem from...lack of beds, insurance payments? Is this d/c from acute care, subacute care, inpatient rehabilitation?

The sentence was revised as follows in page 10: "Bed shortages (30,35,38,41,45) resulting in the pressure to discharge patients in hospitals precluded rehabilitation and delayed post-discharge rehabilitation (31,35,46,47)."

14.-review of 168 patient records...Acute care records? inpatient rehabilitation records? skilled nursing facility records?

The following was added to the text in page 10 "168 stroke patient acute care records".

15. Quality of care - 5 rehab sessions where? In acute care?

We have added the following to the text in page 11 "five rehabilitation sessions during acute care in hospital..."

16.-what is a specialized rehab centre—is this inpatient?

The following was added for clarity in page 11 "specialised sub-acute, in-patient Rehabilitation Centre"

17. Perceptions of care -positive and negative attitudes about what?

Perceptions of care relate to a positive or negative perception of care.

18.-Dissatisfied with healthcare services where...in the hospital? In general? The whole continuum?

The following was added for clarity in page 11 "dissatisfied with the healthcare service along the entire continuum of care, which was"

19.-Line 18 is confusing. Doctors' understanding of what?

This relates to the doctors' understanding of effectiveness of their communication and we have amended to read in page 11: between doctors' understanding, and patients' perception, of the effectiveness of the doctors' communication;

20.Resources - Human resources -line 51: negatively impacted?

Thank you, we have amended the text on page 14: "Mandizvidza (2017) (38) reported that nursing shortage at all healthcare levels in rural KwaZulu Natal negatively impacted basic stroke care."

21.Context -Cultural beliefs and health literacy.-line 29: care seek ability of communities...meaning communities of people with these beliefs combined with poor health literacy?

We have revised the sentence to provide clarity on page 16: "Poor health literacy (60,66,68) and these beliefs further affected the care seeking ability of communities"

22-Discussion - page 19 paragraph about reported supporting factors: Here the authors list many supporting factors that were not included in the Results section. This suggests a biased presentation of the results and calls into question the integrity of this scoping review. The authors' first objective was to describe factors that support and guide achieving universal stroke care in SA, yet the supports are not included in the Results section. Also, Figure 3: Limiting and supporting factors towards achieving UHC, only includes the limiting factors. A scoping review should present ALL of the evidence

We have included a more nuanced approach in the results and the conclusion of the abstract on page 2-3.

Results

"Fifty-nine articles were included in the review. Over half (n=31, 51.6%) were conducted in Western Cape province and most (n=41, 68.3%) were conducted in urban areas. Studies evaluated a diverse range of health system categories and various outcomes. The most common reported component was service delivery (n=47, 76.6%), and only four studies (6.6%) evaluated governance and regulation. Service delivery factors for stroke care were frequently reported as poor and compounded by context related limiting factors. Governance and regulations for stroke care in terms of government support, investment in policy, treatment guidelines, resource distribution and commitment to evidence-based solutions were limited. Promising supporting factors included adequately equipped and staffed urban tertiary facilities, the emergence of stroke units, prompt assessment by health professionals, positive staff attitudes and care, two clinical care guidelines and educational and information resources being available."

Conclusion

This review fills a gap in the literature by providing the range of opportunities and challenges to achieve health for all PWS in SA. It highlights some health system areas that show encouraging trends including to improve service delivery including comprehensiveness, quality and perceptions of care

Factors that support and guide achieving universal stroke care in SA are included in table 2. We have edited the table to clarify the Facilitators and Barriers (p13-15) and highlighted the two areas where no facilitators were reported. In addition, we have highlighted the facilitators more in the results section by including this information in text:

Page 7: We undertook a narrative synthesis of the findings, highlighting facilitators and barriers to achieving health for all PWS in SA. The range of opportunities and challenges to achieve health for all PWS in SA was synthesised and included in the framework diagram.

In addition, we have included supporting factors in Figure 3.

Reviewer 2

Abstract:

1. Please define UHC, PWS, WHO when first used in the abstract. This may be all people read. We have defined UHC, PWS and WHO when first used in the abstract as suggested.

Introduction:

2. Please define all abbreviations when first used in the paper (UHC, PWS, SA)

As advised, we have defined all abbreviations such as UHC, PWS and SA when first used in the paper.

3. Given that this is an international journal, you need to provide a short paragraph on the health system in South Africa in the context of a federal political structure and who is responsible for the services you are covering in your review.

We have included this information as suggested in the introduction in page 4:

South Africa's Constitution guarantees every citizen to have access to health services (section 27 of the Bill of Rights). The SA health system comprises the public sector (government managed) and the private sector. Public health services are divided into primary, secondary and tertiary institutions managed by provincial Departments of Health, with the National Ministry of Health being responsible for policy development and coordination (11). Individuals can access either public or private health services, with access to private health dependant on an individual's ability to pay for services. The majority of South Africans (84%), access health services through government-run public clinics and hospitals (12). SA, stroke care, including rehabilitation, occurs across a range of settings, from tertiary hospitals to remote community primary healthcare facilities and can be provided individually or in a group setting, at home, in a community environment or a specialist centre (2). Whilst public health policy in SA ascribes to primary health care and a decentralised approach, many stroke care and rehabilitation services remain centralised at district and specialist rehabilitation hospitals (13).

4. Fig 1: Please provide a more meaningful title that makes it clear to a reader who may not cover all the content in the text. What is the framework of/ for? Also acknowledge the two frameworks that this one draws on:

Eg Source: this framework incorporates components from the XXX framework (ref) and the yyyy framework (ref)

We have amended the title for figure 1 to more accurately reflect the information contained within the figure, and we have acknowledged the two sources in page 6

Figure 1: Components of the analytical framework that incorporates components from the Health Systems Dynamics Framework (19) and WHO Framework on integrated people-centred health services (20).

5. Search strategy: I know you have stated this in the protocol paper, but I would have liked to see the terms you searched on. You also have not indicated the years that you covered. It is annoying to have to go to the protocol paper for such basic information.

Thank you for your suggestion. We have included our search terms in Supplementary file 1 and note in our methods section on page 6 'An example of the search strategy is available in Supplementary file (S1)'

6. You mention that you have searched to government websites; I would have liked some information about the number, distribution and service components of stroke units and, if possible, health professionals involved in stroke services around the country. This is part of the broader stroke context. Given that you did not go beyond accessing research papers/dissertations and opinions/commentary, you should mention this as a limitation of your review.

We have included the following as a limitation of our review (page 22): "We included research articles, dissertations and commentaries, and there may be evidence missed from health or government websites."

7. Figure 2 is unreadable.

We have provided a higher resolution of Figure 2.

8. Table 1: this is not very informative and is unidimensional. The framework components do not fit into the layout of the table as a whole. I suggest that you create a matrix (this will need to be in landscape), with the framework components each having a column and the other variables each having their rows. You can consolidate the 4 provinces with no records into one. Change 'Area' to 'Area type'

I am not sure of what your crosses mean, but think you can put actual numbers of papers covering that component in each variable category. eg It could be that in WC the studies cover multiple

components so that you can't total them in a row. So make the total for the column. Think about how this table can help readers get an overview of the papers and what they covered.

Please help the reader by putting the legend explaining the abbreviations in the same order as the components in the table.

We have simplified the table to include the characteristics of the included records and provide information on the framework components in the text on page 9.

"Twenty-one articles (35.5%) reported on a single framework component (Service Delivery: n=12; Community Engagement: n=4; Governance and Regulations=2; Context: n=2; Re-Organisation of Care: n= 1) and the majority of articles reported on a combination of two or more framework components (n=38, 64.4%). Twenty-four articles (40.6%) reported on a combination of two framework components (Context and Service Delivery: n =11; Resources and Service Delivery: n=5; Community Engagement and Service Delivery: n =4; Re-Organisation of Care and Service Delivery: n=2; Community Engagement and Re-Organisation of Care: n =2) and fourteen articles (23.7%) reported on three or more framework component combinations (Community Engagement, Context, Service Delivery: n=3; Context, Resources, Service Delivery: n= 2; Community Engagement, Resources, Service Delivery: n =2; Governance and Regulations, Resources, Service Delivery: n=1; Context, Governance and Regulations, Resources n=1; Community Engagement, Context, Resources: n=1; Community Engagement, Context, Resources, Service Delivery: n=4).. "

9. Please explain what you mean by 'doctor-centric'. It can mean different things to different people. Do you mean overemphasis on biomedical or just to suit the doctors' schedules or what? And explain why is this a problem?

We have included the following to clarify the meaning of a doctor-centric model in page 10: "In addition, doctor-led models of care were reported to lead to delays as staff wait for instruction or referral from a doctor before conducting investigations or administering treatment (31, 37).

10. pg 11 Line18. Sentence not understandable – rephrase. Table 2: rephrase title?

We have rephrased the title to Supportive and limiting factors influencing different components of service delivery.

11. Put the service delivery column on the left and the papers from which you derived the evidence on the right. This new column on the right should have a column heading 'Source of evidence: Author (year)

We have adjusted the service delivery column to be placed on the left, and the papers from which we derived the evidence on the right. We have included the column heading 'Source of evidence: Author (year). In addition, we have also clarified which components are facilitators and barriers by creating a third column.

12. Write out MDT in full.

We have written out MDT in full in Table 2 to read: multi-disciplinary team

12. Pg 14, Line 3: information systems are part of infrastructure, not human resources.

We have removed information systems and to highlight that this third component of resources (infrastructure, human resources, financial allocation) was not reported on, we have included the sentence: "There were no articles that reported on the financial allocations in place as a resource for stroke care."

13. Should there be a section or more mention of financial resourcing? Even if in the intro under health system structure and funding.?

We have included the structure of the health system, including financial resourcing in the introduction.

14. Table 3: similar changes to Table 3 in terms of layout and title.

We have changed Table 3 to a similar layout to Table 2

15. Discussion:

Pg 21; line 16.

Start limitations as new paragraph. Or else change the sentence to improve flow.

We have improved the flow of the sentences to include: The framework also acknowledged the social, economic, political context and determinants of health. However, this review has several limitations.

16. Replace 'There was no limitation on study design...' to 'There was no restriction on study design...'

We have edited the sentence as suggested.

17. Pg 21, line 30: no need to redefine WHO if you have done so previously in the paper.

We have removed the redefinition of WHO from the conclusion

18. Is there room somewhere for a statement on primary prevention of stroke (I accept that care poststroke is covered by your paper)?

Given the extensive nature of the scoping review on care post stroke, there is limited room to include a statement on primary prevention of stroke.

VERSION 2 – REVIEW

REVIEWER	Osborne, Candice The University of Texas Southwestern Medical Center
REVIEW RETURNED	24-Aug-2021

GENERAL COMMENTS	A good paper. Your responses are thorough and provided the clarity needed.
--

REVIEWER	Katzenellenbogen, Judith University of Western Australia, School of Population and Global Health
REVIEW RETURNED	12-Sep-2021

GENERAL COMMENTS	The manuscript is much improved. Figure 1 Title: reword please The title is a bit strange. You need to link it to your study. How about: Analytic framework for health system-related factors that limit or support Universal health Coverage, incorporating components from the Health Systems Dynamics Framework () and WHO Framework on Integrated People-Centred health Services () What is the bit about identifying the research question? ? leave off maybe? Table 1: The paragraph you have added is difficult to read. One never gets a good, multidimensional overview of the papers you have reviewed. Many long lists of factors through the manuscript. You may try to somewhat more meaningful sentences.
---

VERSION 2 – AUTHOR RESPONSE

Reviewer: 1 Comments to the Author:

A good paper. Your responses are thorough and provided the clarity needed.

Thank you for your review.

Reviewer: 2 Comments to the Author:

The manuscript is much improved.

Thank you for your review and further suggestions

Figure 1 Title: reword please

The title is a bit strange. You need to link it to your study. How about: Analytic framework for health system-related factors that limit or support Universal health Coverage, incorporating components from the Health Systems Dynamics Framework () and WHO Framework on Integrated People-Centred health Services ()

Thank you for your suggestion. We have re-titled Figure 1 from “Components of the analytical framework that incorporates components from the Health Systems Dynamics Framework (19) and WHO Framework on integrated people-centred health services (20)” to: “Analytic framework for health system-related factors that limit or support UHC, incorporating components from the Health Systems Dynamics Framework (22) and WHO Framework on Integrated People-Centred health Services (23)

What is the bit about identifying the research question? ? leave off maybe?

We have bolded 'Identifying the research question' and formatted as a new paragraph to ensure that it is read as a subheading.

Table 1: The paragraph you have added is difficult to read. One never gets a good, multidimensional overview of the papers you have reviewed.

We have edited the paragraph following Table 1 to provide a multi-dimensional overview as follows:

Twenty-one articles (35.5%) reported on a single framework component, of which service delivery (n=12/21, 57.1%) was the most commonly described. The majority of articles included a combination of components (n=38, 64.4%); 24 articles (40.6%) reported on two framework components, and fourteen articles (23.7%) reported on three or more. Of the combination of components, Context was most commonly combined with Service Delivery (n = 11/38, 28.9%) followed by Resources and Service Delivery (n=5/38, 13.1%).

Many long lists of factors through the manuscript. You may try to somewhat more meaningful sentences.

We have curated our language as follows:

P10 Long waiting times contributes to the paucity of therapy sessions... Consequently, delays in investigations being were found to be associated with a significant increase in length of stay (42) ...

P11 Bed shortages (30,35,38,41,45) resulted in the pressure to discharge patients from hospitals, which precluded rehabilitation

P11 There was conflicting evidence regarding perceptions of care. Ten studies reporting positive staff attitudes

P13 – 14 Table 2: we have edited the table to create more meaningful sentences

P14 Furthermore, more specialised services often remained inaccessible (30,31,45) as their geographic location required even longer travel times..

P15 Financial burden was found to increase when spouses became primary caregivers (without gainful employment) or through the employment of additional caregivers (57). Costs post-stroke were

high due to additional caregiving expenses (60,73) and studies found that there was limited access to disability-, old age- or child-support grants (52,65). The financial burden among rural stroke survivors was compounded by low income before the stroke,

P18 Governance and Regulations were the most limited component reported, which demonstrates a deficit in leadership and policy for how stroke care should be implemented and conducted at all levels of care

P19 Included articles evaluated a diverse range of health system categories and various outcomes, with the majority of studies reporting on two or more framework components. There were several key limiting factors toward achieving UHC, which included a lack of governmental regulation in terms of stroke policies and guidelines poor timeliness of care, a lack of the continuity of care and a lack of a comprehensive multi-disciplinary team at rural health facilities. Furthermore, bed and staff shortages and a lack of stroke-specific training, poor access to acute care and diagnostic equipment contributed to limiting UHC. Regular medication stockouts, lack of caregiver training and contradictory reports on perceptions of care were also found to be limiting factors..... There were also many supporting factors toward achieving UHC for PWS in SA, which included adequately equipped and staffed urban tertiary facilities, the emergence of Stroke Units in urban... Resources that were available to support achieving UHC include two clinical care guidelines, and educational and information resources being available online.

We have also removed repetition in the discussion section.

For example:

P20 The main hindrances affecting service delivery in SA related to training, resources and communication channels. Poor referral networks and few rural rehabilitation facilities were compounded by inadequate caregiver training, lack of stroke-specific staff training, bed shortages, and diagnostic equipment. As a result, many PWS are lost to follow-up care leading to poor management of comorbidities and potentially placing patients at risk of recurrence and secondary

VERSION 3 – REVIEW

REVIEWER	Katzenellenbogen, Judith University of Western Australia, School of Population and Global Health
REVIEW RETURNED	28-Oct-2021
GENERAL COMMENTS	Accept